# Evaluation of the accuracy of the IDvet serological test for Mycoplasma bovis infection in cattle using latent class analysis of paired serum ELISA and quantitative real-time PCR on tonsillar swabs sampled at slaughter

Nelly Marquetoux[1]*, Matthieu Vignes[2], Amy Burroughs[3], Emma Sumner[3], Kate Sawford[3,4], Geoff Jones[2]

1 EpiCentre, School of Veterinary Science, Massey University, Palmerston North, New Zealand, 2 School of Mathematical and Computational Sciences, Massey University, Palmerston North, New Zealand, 3 Ministry for Primary Industries New Zealand, Wellington, New Zealand, 4 Kate Sawford Epidemiol Consulting, Callala Bay, NSW, Australia

* n.marquetoux@massey.ac.nz

## Abstract

*Mycoplasma bovis* (Mbovis) was first detected in cattle in New Zealand (NZ) in July 2017. To prevent further spread, NZ launched a world-first National Eradication Programme in May 2018. Existing diagnostic tests for Mbovis have been applied in countries where Mbovis is endemic, for detecting infection following outbreaks of clinical disease. Diagnostic test evaluation (DTE) under NZ conditions was thus required to inform the Programme. We used Bayesian Latent Class Analysis on paired serum ELISA (ID Screen Mycoplasma bovis Indirect from IDvet) and tonsillar swabs (qPCR) for DTE in the absence of a gold standard. Tested samples were collected at slaughter between June 2018 and November 2019, from infected herds depopulated by the Programme. A first set of models evaluated the detection of active infection, i.e. the presence of Mbovis in the host. At a modified serology positivity threshold of $SP\% > = 90$, estimates of animal-level ELISA sensitivity was 72.8% (95% credible interval 68.5%—77.4%), respectively 97.7% (95% credible interval 97.3%—98.1%) for specificity, while the qPCR sensitivity was 45.2% (95% credible interval 41.0%—49.8%), respectively 99.6% (95% credible interval 99.4%—99.8%) for specificity. In a second set of models, prior information about ELISA specificity was obtained from the National Beef Cattle Surveillance Programme, a population theoretically free—or very low prevalence—of Mbovis. These analyses aimed to evaluate the accuracy of the ELISA test targeting prior exposure to Mbovis, rather than active infection. The specificity of the ELISA for detecting exposure to Mbovis was 99.9% (95% credible interval 99.7%—100.0%), hence near perfect at the threshold SP%=90. This specificity estimate, considerably higher than in the first set of models, was equivalent to the manufacturer's estimate. The corresponding ELISA sensitivity estimate was 66.0% (95% credible interval 62.7%-70.7%). These results confirm that

**Data Availability Statement:** All relevant data will be provided in the form of contingency tables as Supporting information files.

**Funding:** This work was funded by the Strategic Science Advisory Group on Mycoplasma bovis, Ministry for Primary Industries, New Zealand (https://www.mpi.govt.nz/biosecurity/mycoplasma-bovis/strategic-science-advisory-group/). The funders had no role in study design, data collection and analysis, decision to publish, or preparation of the manuscript.

**Competing interests:** The authors have declared that no competing interests exist.

the IDvet ELISA test is an appropriate tool for determining exposure and infection status of herds, both to delimit and confirm the absence of Mbovis.

## Introduction

Mycoplasma bovis (Mbovis) is a bacterium of the Mollicute class occurring sporadically in most dairy cattle systems worldwide [1], including in pasture-based production systems such as Australia [2]. Mbovis is potentially pathogenic for cattle, associated mostly with respiratory diseases in calves or feedlot cattle, and mastitis in dairy cows. The bacteria colonize mucosal surfaces, primarily the upper respiratory tract, as well as the mammary gland in dairy cows [3]. Highly contagious in nature, the transmission of Mbovis within the herd occurs through aerosols via direct contact between individuals, or by feeding calves infected milk [1, 4]. Transmission can also occur between lactating cows during milking [5]. Chronic and asymptomatic infections can occur for extended periods of time, up to several years [3]. Intermittent shedding can occur from various mucosal sites [3, 6], which poses a challenge for accurate diagnosis at the individual animal level and for biosecurity purposes [7]. Intermittent shedding from the mammary gland of sub-clinically infected dairy cows can persist through successive lactation periods [8]. In calves, Mbovis long-term survival in the host has been related to mechanisms to avoid phagocytosis [9]. Persistence in the host as asymptomatic carriage could promote continued circulation in the herd and recurring outbreaks of mastitis [1]. Chronic infection with intermittent shedding thus appears critical in the long-term maintenance in a herd [3]. On the other hand, clearance of infection has also been observed from individual animals [10] and from previously infected herds [4]. Introduction into a naive herd is usually the result of importing infected cattle [11].

In New Zealand (NZ), Mbovis was first detected in July 2017. A national response was launched in July 2017 to delimit, contain and ultimately eliminate the infection from NZ. Accurate classification of herds as infected versus not infected is paramount to the success of the Mbovis National Eradication Programme (subsequently called the Programme). Diagnostic test evaluation (DTE) represents an important milestone for informing herd-level surveillance to achieve elimination.

Testing for the Programme consists of (1) on-farm screening using serum ELISA in live animals in farms at risk of infection and (2) samples collected from animals sent to slaughter during Programme depopulation, with individual serum ELISA and tonsillar swabs to conduct quantitative PCR (qPCR). National background surveillance also occurs in the general beef population and via testing of milk collected by dairy companies, as well as notifications from field veterinarians in case of a clinical suspicion. All diagnostic tests are interpreted at the herd level with a view to detect sub-clinical infection, prevent transmission between herds by removing all infected herds and eliminate the organism from NZ. Diagnostic tests for Mbovis were developed and evaluated in other countries, in a context of endemic infection, to confirm Mbovis aetiology in the face of clinical disease [12]. By contrast, in New Zealand, these tests are mostly applied to screen a population of animals for infection irrespective of the presence of clinical signs, thus requiring evaluation in this specific context.

PCR is not an appropriate gold standard to evaluate the ELISA test, due to the intermittence of shedding and other factors that affect diagnostic sensitivity [7]. Latent class analysis (LCA) provides a method of reference for DTE in the absence of a gold standard and can provide sensitivity and specificity estimates for both serum ELISA and the qPCR test on tonsillar swabs [13]. The principle is to model the observed biomarker responses of individuals based on the unobserved (i.e. latent) infection status of individuals. We used the Hui-Walter modelling

approach, which consists of splitting the study population into several subpopulations with distinct prevalences to achieve identifiability. Identifiability is important to ensure that estimates of test accuracy do not rely as heavily on prior values as would be the case with unidentifiable Bayesian models [28].

The presence of antibody titers in serum detects infection with Mbovis and, more generally, past exposure to Mbovis. The presence of detectable bacterial DNA on a tonsillar swab measures local harbouring of Mbovis in the palatine tonsillar crypts, or respiratory secretion. Thus, consideration must be given to the target condition when serology and qPCR results are analysed together. Standard DTE ideally requires prospective random sampling from the population targeted for sampling and a control population, and testing by two or more tests. Here we used surveillance data instead, collected purposively with the aim to detect infection by the Mbovis Eradication Programme. We implemented LCA using Bayesian inference [14].

This study presents a descriptive analysis of serology conducted by the Mbovis eradication Programme between June 2018 and Novememeber 2019, and the outcomes of LCA conducted on paired serology and PCR data collected by the Programme at slaughter.

## Materials and methods

The STARD-BLCM standards were followed for this study [13]. In the context of the Programme, the purpose of testing is to identify and remove all herds infected with Mbovis, starting in herds with a known epidemiological risk event of exposure to Mbovis, with a view to eradicate the organism from the NZ cattle population. The tests are applied to a population irrespective of the presence of clinical signs, with low animal- and farm-level prevalence among the tested population. In the delimiting phase, active surveillance by the Programme is targeting infection in a population identified as high-risk. Over time, as the prevalence of infection decreases, the importance of active surveillance in the general cattle population (low risk of infection) will prevail, to detect last pockets of infection and prove freedom. For that purpose, the target condition of testing at the animal-level is "exposure to Mbovis" (i.e. current or past infection). Our study used two distinct populations and sets of models to assess the test performance in relation with two distinct target conditions, to address the purposes of the test adequately.

### Study populations

**Delimiting population, Mbovis National Eradication Programme.**  The Delimiting population is the population of farms tested by the Programme on the basis of the identification of a known risk event—usually known cattle movements traced to or from other farms confirmed as infected. All trace cattle from known infected farms are sent to slaughter and tested by qPCR on tonsillar swab as well as serum ELISA, as part of determining the infection status of receiving farms. Moreoever, at risk farms undergo herd-level ELISA testing, consisting of two rounds of sampling a few weeks apart (median 4.3 weeks, 95% quantile 2.9—13.4 weeks). All cattle traced to or from infected properties were sampled. Additionally, in at-risk groups of animals (entire farms, or groups of animals in contact with trace animals), a random subsample of cattle were sampled up to the required sample size. The sample size per group of animal sampled changed in 2019 from 100 individuals per group per round, to a minimum of 220 individuals, or all the animals in groups smaller than the minimum sample size. In some instances, subsequent rounds of herd-level serology followed the initial two rounds to determine the infection status of the sampled groups and farm. Infection was confirmed if 3% or more of the animals are ELISA positive above a threshold of SP%=90 in any two rounds of ELISA screening, or following a single positive qPCR result with sequencing of Mbovis. If

group-level infection was confirmed, all groups found to be infected and any other groups at higher risk of infection were sent to slaughter by the Programme as part of a legal depopulation, where a subset of the depopulated groups (i.e. previous reactors from on-farm serology) were tested by ELISA and qPCR on tonsillar swabs.

The slaughter population thus represented the subset of data from the delimiting population (targeted, high risk individuals) with paired ELISA and qPCR results available at the individual-level. This slaughter subset was used for all LCA. ELISA and qPCR testing of the slaughter population was carried out by MPI's Animal Health Laboratory (AHL) during the timespan covered by the dataset (sample collection date ranging from 09 June 2018 to 27 November 2019).

**Background National Beef Cattle Surveillance.** Background national beef cattle surveillance started in 2019, screening the general beef cattle population of NZ, to determine the prevalence of Mbovis infection in the beef industry. This programme consists of ongoing monitoring of cattle arriving at a large aggregator, beef breeding herds and cattle at slaughter. This population represents a population likely uninfected, or with a very low between-herd prevalence of Mbovis. Only serology results are available for this stream of surveillance.

To summarise, the source population was the slaughter population (high risk subset of the Delimiting population). In a first set of models, the target population was the Delimiting population (at risk population linked to known infection, population screened by the Programme). In a second set of models, the target population was the wider general population of cattle, represented by cattle surveyed in the National beef cattle surveillance.

## Diagnostic tests and latent condition

- **serum ELISA**: Serum samples were tested using the ID Screen Mycoplasma bovis Indirect produced by IDvet. The ELISA was performed as described by the manufacturer but using the sample-to-positive ratio (SP%) cut point described below, to categorise results into "positive" or "negative". The positivity threshold recommended by the manufacturer was SP% =60, whereas the Programme used a higher "epidemiological" threshold of SP%=90 to reduce false positive test outcomes at the individual sample level. Series of models were fitted with thresholds varying from 60 to 100 to analyse the sensitivity to the threshold.

- **qPCR on tonsillar swabs**: the qPCR testing was carried using the LSI VetMAX Mycoplasma bovis kit (Life Technologies New Zealand Ltd.) as described by [15]. The qPCR test results were reported as "positive" (or "detected", recoded as "positive") with a Ct value of 38 as the positivity threshold, "negative" (or "undetected", recoded as "negative") or "SEE COMMENT". The latter occurred when the result could not easily be dichotomised by the laboratory. In this case, a free text comment was available at the accession level, summarising molecular diagnostics for all samples in the accession, thus precluding a systematic treatment of this information at the sample level. There were a total of 117 "SEE COMMENT" results, compared to 1609 "positive" and 57320 "negative" results. The comments were qualitatively analysed and fell mostly into two categories:

  1. late amplification or not all replicates gave rise to amplification (usually related to late amplification). This category thus represents the "weak positive", results too close to the positivity threshold to unambiguously categorise them as "positive".

  2. the qPCR result was positive but the amplification products were further analysed by conventional PCR and sequencing, which did not confirm the molecular diagnosis of Mbovis, or were inconclusive. Importantly, conventional PCR and sequencing were

performed routinely on a small subset of samples positive by qPCR collected from each farm early on, and were not systematically performed for all subsequent qPCR-positive samples from that infected farm, once Mbovis infection was confirmed by sequencing. For all additional positive samples from an infected farm that were not followed by conventional PCR, the laboratory reporting process would be to report this same result as "positive". This category of "SEE COMMENT" results would thus represent false positive qPCR results.

Rather than ignoring these results, we chose to include them in the analysis as a third PCR category "C".

- **Definition of the latent condition for the LCA** The latent condition modelled jointly by ELISA and qPCR testing at slaughter is determined by the overlap of the distribution of biomarkers detected by either test in the host. In this case, it can be defined as "active infection with Mbovis"—including latent and sub-clinical infection. Active infection results from the primary colonisation of mucosal surfaces so as to trigger the immune response. The host currently harbours the bacteria, with potential for intermittent shedding that can result in spreading of the agent. In other words, the latent condition captured when using both ELISA and qPCR is the entire duration during which an animal has Mbovis somewhere in its tissues. Tonsillar swab qPCR might not detect active infection due to intermittent shedding, variable colonisation sites or poor sampling technique. Animals exposed to Mbovis but without current infection might not be captured by this latent condition as they do not harbour the agent in their tissues. A diagrammatic representation of the latent condition is depicted in Fig 1.

## Description of test results

We estimated the apparent seroprevalence in all sampling rounds from the Programme as well as for National Beef Cattle Surveillance. We then examined the distribution of within-herd positive ELISA results in the first round of sampling (initial screening). This initial screening is mandatory for all at-risk herds entering the Programme and thus representative of the population at risk targeted by the Eradication Programme. The overall true prevalence of infection in the delimiting population is expected to be low, considering the very high proportion of negative test results in negative herds as well as a fair proportion of negative animals in most positive herds. We used the ELISA kit positivity threshold of $SP\% = 60$ for these analyses.

Further, to study the correlation of titres over time in the same animal, we identified individuals identified by their unique electronic identifier and appearing in two subsequent rounds of sampling (n = 87116) to describe the agreement in titres, using the Cohen's Kappa coefficient.

## Prior knowledge about diagnostic tests in use in the Programme and animal-level prevalence of Mbovis

**ELISA sensitivity and specificity.** From the manufacturer's kit insert, the analytical specificity of the IDvet ELISA was virtually perfect (CI 96.3%—100.0%, n = 100). The sensitivity was estimated to 95.7% (CI 87.3%- 100.0%) based on 23 calves with clinical signs of disease and using Western Blot as a gold standard (ID Screen[R] Mycoplasma bovis Indirect, kit insert). The IDvet ELISA was also evaluated against other serological tests using LCA in one interlaboratory validation trial [16]. To inform sensitivity, sera were sourced that had "previously been determined as positive for M. bovis", from clinically affected animals in Mbovis infected

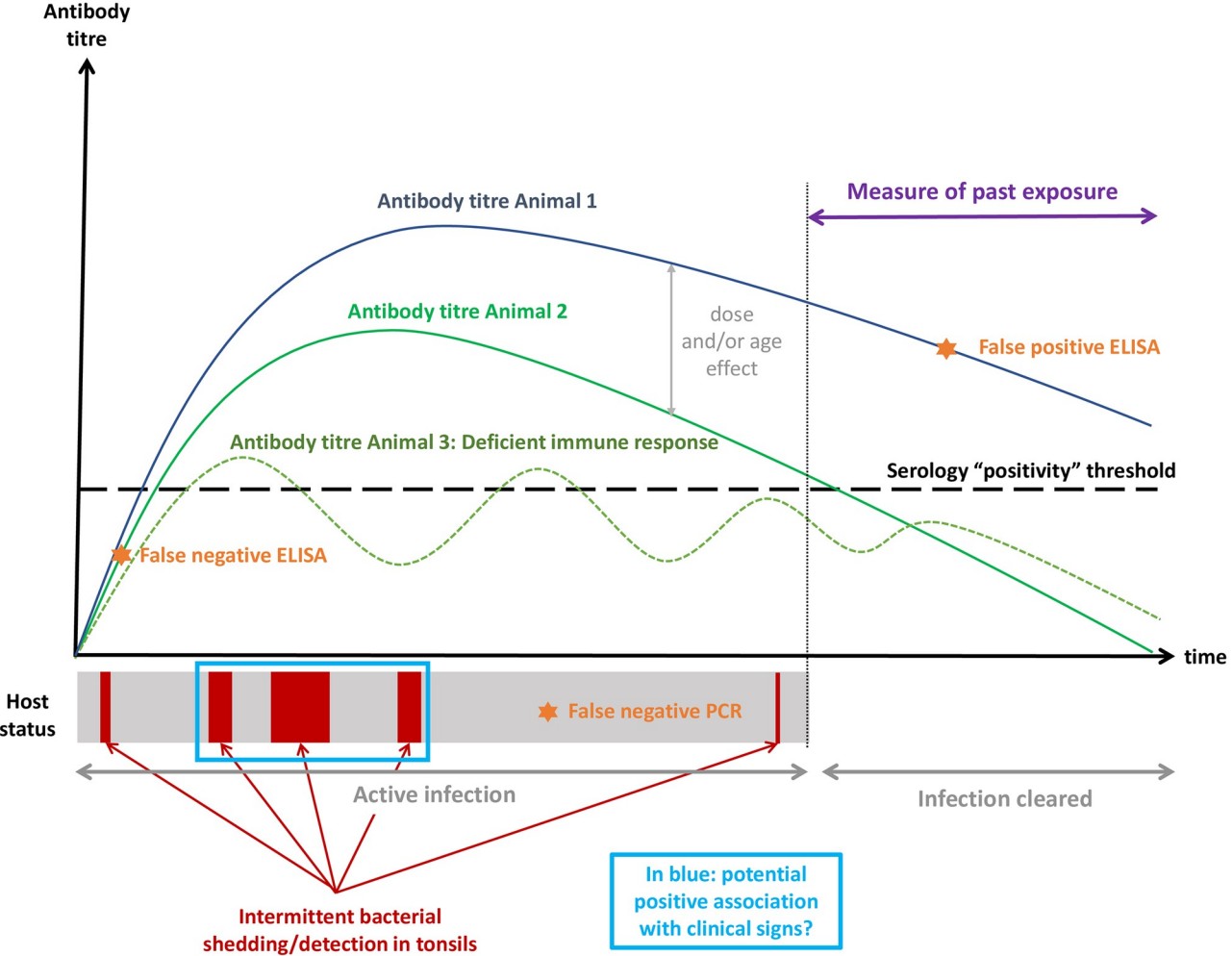

**Fig 1. Diagrammatic representation of Mbovis physiopathology and possible distribution of biomarkers associated with ELISA and qPCR testing, with corresponding latent condition for LCA.**

herds, and sera from uninfected herds from a region free of Mbovis were collected to inform specificity. The estimated sensitivity was 93.5% and the estimated specificity was 98.6%. The sensitivity estimate obtained in this study, given the choice of samples, pertains to confirmation of clinical disease caused by Mbovis in heavily infected herds; this is irrelevant in relation to detecting infection in clinically healthy herds in a low prevalence context.

**qPCR sensitivity and specificity.** Laboratory validation of the qPCR kit (TaqVet™, Life-technologies) by the manufacturer demonstrated the absence of cross-reaction towards all other related pathogens included in the study, with a limit of detection of 10 DNA copies per reaction [17]. Despite its high analytical sensitivity when Mbovis DNA is present in the sample, the diagnostic sensitivity of the PCR is hampered by intermittent shedding, latent infection, variable colonisation sites, and poor sampling technique [4]. qPCR was therefore deemed unsuitable as a screening test for prevalence studies for Mbovis [2]. In one study [7], fifteen out of sixteen cows with recent clinical Mbovis mastitis had seroconverted, but bacterial isolation by culture was only achieved in three vaginal swabs, and none of the nasal swabs.

However, no tonsillar swabs were collected in this study. In another herd with a recent outbreak of Mbovis mastitis, detectable colonisation from multiple mucosal sites was monitored at quarterly intervals for a year. The proportion of positive samples varied from 0 to 47%, but was below 10% in most sampling events [10]. Colonisation of multiple mucosal sites was relatively common and nasal swabs were the preferential site for bacterial isolation, although tonsillar swabs were not collected. Colonisation of mucosal sites in asymptomatic carriers was detectable in some animals for over a year and was associated with secondary infection of replacement stock, as per our definition of active infection. Conversely, complete clearance from all sampled mucosal sites was observed in other animals previously colonised [10], supporting the hypothesis that some animals are able to recover from active infection. Internally, the AHL provisionally estimated the overall diagnostic sensitivity of the qPCR test on tonsillar swabs at 30%, from early data collected for the Programme (Doug Begg Pers. Comm.). However, little is known about the dynamics of humoral immune response and shedding during Mbovis infection and neither test has been validated for the purpose and target condition for which they are used in NZ. Therefore, well informed prior knowledge about the sensitivity and specificity of qPCR was scarcely available.

Based on the above and considering the large volume of datapoints available for analysis and the fact that all models were identifiable, we used diffuse (weak) albeit biologically plausible Beta distribution priors for the sensitivity and specificity of both tests (Table 1).

**True prevalence.** The within-herd prevalence of infection appears highly variable from herd to herd. There are reports of nasal infection in 6% of dairy calves in clinically healthy herds, others of less than 7% in "non-stressed" beef calves [3]. However, a very high animal-level prevalence can occur in herds with clinical disease, or feedlots and calf rearing situations [3]. Seroconversion of up to 100% of feedlot cattle has been observed within six weeks of the introduction of infected cattle [2]. High seroconversion levels were similarly observed in NZ during a longitudinal study of one persistently infected aggregator, in weeks following the introduction of naive cattle. The proportion of animals positive by qPCR in tonsillar swabs at slaughter was also very high (Comm. Pers. Olivia Kingston), confirming that both tests are markers of active infection. Nevertheless, the overall animal-level apparent seroprevalence in animals tested by the Programme is very low: 3% of seropositivity overall across all sampling events and all farms tested. We used a diffuse Beta prior distribution for the true prevalence of infection, compatible with the above information (Table 1). The same prevalence prior was used for all sub-populations, thus incorporating adequate uncertainty to encompass higher levels of prevalences in some subgroups.

Let us denote $Cp$ the probability $Pr(PCR = C|\text{infected})$ and $Cn$ the probability $Pr(PCR = C|\text{non-infected})$. Given the lack of prior knowledge about the probabilities of qPCR results reported as comments by the laboratory, flat beta(1,1) priors were used for these two parameters in all analyses.

Sensitivity analysis for the effect of priors was carried out, using alternative priors values as well as flat distributions for all priors.

**Table 1. Parameters of prior Beta distributions for the main analyses.**

| Parameter | shape 1 | shape 2 | Probability |
|---|---|---|---|
| ELISA sensitivity | 1.57 | 1.14 | 50% sure>0.6, Mode 0.8 |
| qPCR sensitivity | 2.03 | 2.55 | 75% sure<0.6, Mode 0.4 |
| ELISA specificity | 68.8 | 4.6 | 80% sure>0.8, Mode 0.95 |
| qPCR specificity | 20.5 | 1.4 | 80% sure>0.9, Mode 0.98 |
| True prevalence | 1.65 | 2.53 | 90% sure<0.7, Mode 0.3 |

## Partitioning the slaughter population into sub-populations of distinct prevalences

The slaughter dataset is a purposive sample with a relatively homogeneous and high baseline risk of infection, further skewed—by design—towards farms with high within-herd prevalence. There was no available contrasting population tested by qPCR. Nevertheless, the livestock population in NZ is naturally highly clustered by the physical separation between the North and South Island, with relatively few livestock movements across islands in comparison to the frequency of movements within island [18]; this cattle movement pattern determines distinct overall prevalence levels for transmissible infectious diseases [19]. We therefore partitioned the population by cattle from farms in the North versus the South Island.

## LCA Hui-Walter models

We used a series of Hui-Walter models [20], requiring two tests performed independently of each other on each individual, which was the case for all cattle from depopulated groups *i.e.* the slaughter population. The assumptions of these models are that:

- The study population can be sub-divided into groups, defined *a priori*, with distinct prevalences in each;

- The accuracy of both tests is constant across the various populations, i.e. similar distribution of the measured response in infected animals of the various populations;

- The results of the two tests are independent of each other (conditionally on infection status).

Two sub-populations with presumably distinct prevalences were obtained by splitting the slaughter population according to the geographic location of the farm of origin: North versus South Island. For each sub-population $i$ with $i$ taking the value "North Island" or "South Island", $n_i$ cows are jointly tested by ELISA and qPCR. The vector of counts obtained for each test result combination (pos-pos, pos-neg, pos-comment, neg-pos, neg-neg, neg-comment) is denoted $y_i$. Thus, $y_i|p_i \sim$ Multinomial$(n_i, p_i)$, where $p_i$ is the vector of the six probabilities $p_i = (p_{i1}, \ldots, p_{i6})$ associated to the six test result combinations. These probabilities are modelled conditionally on the latent status denoted as $D+$ (presents the latent condition) and $D-$ (does not present the latent condition).

Let $T_1$ refer to ELISA, $T_1+$ and $T_1-$ refer to the outcome of the ELISA test, $T_2$ refer to qPCR, $T_2+$, $T_2-$ and $T_2C$ refer to the outcome of the qPCR test. Let $se_1$ and $sp_1$ denote the sensitivity and specificity of the ELISA, respectively $se_2$ and $sp_2$ for qPCR, and $tp_i$ the true prevalence in island $i$. According to the law of total probabilities, $p_i$ can be modelled as:

$$p_{i1} = Pr(T1+, T2+) = tp_i\, se_1\, se_2 + (1 - tp_i)\,(1 - sp_1)\,(1 - sp_2 - Cn) \tag{1}$$

$$p_{i2} = Pr(T1+, T2-) = tp_i\, se_1\,(1 - se_2 - Cp) + (1 - tp_i)\,(1 - sp_1)\,(sp_2) \tag{2}$$

$$p_{i3} = Pr(T1+, T2C) = tp_i\, se_1\, Cp + (1 - tp_i)\,(1 - sp_1)\, Cn \tag{3}$$

$$p_{i4} = Pr(T1-, T2+) = tp_i\,(1 - se_1)\,(se_2) + (1 - tp_i)\, sp_1\,(1 - sp_2 - Cn) \tag{4}$$

$$p_{i5} = Pr(T1-, T2-) = tp_i\,(1 - se_1)\,(1 - se_2 - Cp) + (1 - tp_i)\, sp_1\,(sp_2) \tag{5}$$

$$p_{i6} = Pr(T1-, T2C) = tp_i\,(1 - se_1)\,Cp + (1 - tp_i)\,sp_1\,Cn \tag{6}$$

**ELISA thresholds.** For each of the model types presented above, we fitted a series of models to explore the effect of varying ELISA thresholds from SP%=60 to SP%=100 by increments of 10 (one for each threshold).

## Parameter estimation and model validation

Bayesian inferences for all models were obtained by numeric approximation of the joint posterior distribution, using the JAGS software (version 4.10; http://mcmc-jags.sourceforge.net/). Samples from the posterior distributions of the parameters were obtained by Markov chain Monte Carlo (MCMC) sampling. For all models, the burn-in period was adjusted by monitoring convergence, using the Gelman-Rubin criterion and a visual inspection of three chains run in parallel. The sample size of the Monte-Carlo Markov chain was adjusted to take into account auto-correlation between iterations, to achieve an effective sample size equivalent to 5000 independent samples for parameters of true prevalence, sensitivity and specificity. Initial values for the parameters were generated by sampling at random from the prior distribution of the parameters. To avoid convergence towards mirror solutions, we included the condition $Se + Sp > 1$ in the initialisation and the prior specification. We evaluated the effect of different sets of initial values and alternative prior distribution, as well as completely diffuse priors Beta (1, 1) for all parameters.

Visual inspection of the variations of parameter estimates with varying ELISA thresholds was used as a diagnostic for model assumptions. Only the accuracy estimates for the ELISA test should vary with varying ELISA thresholds, while accuracy estimates for the qPCR test and true prevalence should remain constant.

## Using complementary ELISA screening data to model exposure to Mbovis via constrained priors

The ELISA test alone, without the paired qPCR test on tonsillar swabs performed at slaughter, measures "past exposure to Mbovis", irrespective of current infection status. This state is distinct from the latent condition previously defined, corresponding to current infection. For the purpose of eliminating Mbovis from the population and demonstrating the absence of disease, exposure or past infection needs to be included in the target condition of interest, as there are no diagnostic tests that can be used to discriminate accurately between current and past infection with Mbovis at the individual or herd level. Additional ELISA data outside of the slaughter data used in the above models may be used to adjust the estimates of these models towards the desired target condition, i.e. exposure to Mbovis with or without current infection. To this avail we used alternative data from National Beef Cattle Surveillance, representing an almost entirely unexposed population. The apparent prevalence (AP) in this population can provide a lower bound for the ELISA specificity estimate towards exposure to Mbovis, denoted $Sp_{exp}$.

Providing that $Se + Sp > 1$, then $Sp >= 1 - AP$. A lower bound for $Sp_{exp}$ in National Beef Cattle Surveillance can thus be calculated as:

$$Sp_{exp} \geq 1 - AP(round1) \tag{7}$$

We used the upper limit of the 95% CI of the apparent prevalence (Table 2) to estimate a robust value for the lower bound of $Sp_{exp}$. We then used this value as strong prior information

**Table 2. Apparent seroprevalence for the slaughter sampling round (Delimiting population) and for National Beef Cattle Surveillance population, at different levels of ELISA SP% thresholds.**

| Elisa threshold | Positive | Negative | n | App. prevalence (%) | CI low (%) | CI up (%) |
|---|---|---|---|---|---|---|
| **Delimiting Slaughter** | | | | | | |
| 60 | 8913 | 90629 | 99542 | 8.95 | 8.78 | 9.13 |
| 70 | 7805 | 91737 | 99542 | 7.84 | 7.68 | 8.01 |
| 80 | 6725 | 92817 | 99542 | 6.76 | 6.60 | 6.91 |
| 90 | 5670 | 93872 | 99542 | 5.70 | 5.55 | 5.84 |
| 100 | 4728 | 94814 | 99542 | 4.75 | 4.62 | 4.88 |
| **National Beef Cattle Surveillance** | | | | | | |
| 60 | 37 | 20137 | 20174 | 0.18 | 0.13 | 0.26 |
| 70 | 30 | 20144 | 20174 | 0.15 | 0.10 | 0.22 |
| 80 | 24 | 20150 | 20174 | 0.12 | 0.08 | 0.18 |
| 90 | 21 | 20153 | 20174 | 0.10 | 0.07 | 0.16 |
| 100 | 17 | 20157 | 20174 | 0.08 | 0.05 | 0.14 |

for the LCA model, representing an *ad hoc* adjustment method to obtain estimates pertaining to "prior exposure" instead of "active infection". More precisely, we used a uniform distribution between the above lower bound for $Sp_{exp}$ and 1, as a prior for ELISA specificity to re-fit the LCA models. We applied this method to models using ELISA thresholds of 60 and 90.

## Supplementary analysis for herd-level interpretation of the ELISA test

We used animal-level ELISA accuracy estimates obtained by LCA to appraise different options for herd-level test interpretation to optimise the surveillance. We calculated herd sensitivity (Hse) and herd specificity (Hsp) for varying numbers of tested animals in the herd (n, range 10–230), varying levels of within-herd true prevalence of infection (TP, range 1%-10%) and varying cut-points (proportion of seropositive animals required to declare the herd as infected, range 1%-10%). We assumed a fixed herd size close to the average dairy herd size in NZ (N = 400), but also evaluated the effect of changing the herd size to N = 300. The hypergeometric distribution was used to calculate Hse and Hsp (sampling without replacement) to take into account finite herd size [21]. The result was expressed as the proportion of herds that would be misclassified for each value of the input parameters, corresponding to Hse and Hsp for one unique sampling event for one herd.

We used two distinct sets of sensitivity and specificity estimates, corresponding to:

- **Scenario 1**: Detection of active infection with Mbovis in the herd, manufacturer's threshold (SP%=60), sensitivity = 0.8719 and specificity = 0.9533.

- **Scenario 2**: Detection of prior exposure to Mbovis in the herd, corrected LCA estimates obtained using data from National Beef Cattle Surveillance, epidemiological threshold (SP% =90), sensitivity = 0.6604 and specificity = 0.9987.

We use the notation CI to denote credible intervals (for the results of the LCA models) or confidence intervals (for other descriptive results) at the 95% confidence level.

Animal ethics approval was not required for this study as it did not require primary data collection. All data used for these analyses were collected by the Ministry for Primary Industries' Mycoplasma bovis Eradication Programme under the Biosecurity Act 1993. An anonymised version was made available to the research group for analyses.

## Results

### Descriptive ELISA results

Using the manufacturer's threshold for seropositivity (SP%=60), the overall apparent seroprevalence in all cattle sampled as part of the Programme (Delimiting population, selecting only the first titre for repeat tests on the same animal over time) was 2.81% (n = 398148, CI 2.76–2.86). Details of apparent seroprevalence for various ELISA positivity thresholds for the Delimiting population and the National Beef Cattle Surveillance population can be seen in Table 2. The number of herds screened in the Delimiting population was 1645.

The apparent seroprevalence in the first on-farm screeening round (n = 238,435) was 1.8% at the kit positivity threshold (SP%=60) and 1.1% at the epidemiological interpretation threshold (SP%=90). The apparent seroprevalence was clustered by herds, with 66% of herds with no positive ELISA results and 10.2% of herds with over 4% (Fig 2). This finding reflects the fact that true infection clusters within herds.

An increase in seroprevalence in successive rounds of the Programme was observed (Fig 3). This reflected the selection bias towards the ELISA positive animals and groups in subsequent confirmatory sampling, leading up to slaughter sampling with the highest apparent prevalence of 10.8%. To some extent it may also account for the change of within-herd prevalence of infection over time between successive sampling rounds and at slaughter. The progression of ELISA titres in subsequent samples from cattle in the Delimiting population sampled twice (n = 87116) is described in Fig 4.

The agreement between two successive titres in the same animal was "moderate" (Kappa value = 0.59, $p < 0.001$), mostly due to the overwhelming proportion of animals negative to both tests. However, among animals positive in at least one of the two rounds (n = 3323), 57% had their titre switch from positive to negative or vice versa, with virtually equiprobability one way or the other. The median time difference between the first two subsequent rounds for the same animal was 5 weeks (95% quantile 3—18.7 weeks).

On the other hand, the apparent seroprevalence for National Beef Cattle Surveillance (n = 20,174) was 0.18% at the kit positivity threshold (SP%=60) and 0.10% at the epidemiological interpretation threshold (SP%=90). The target population can thus be considered as virtually uninfected and unexposed.

### Hui-Walter models results

In the Delimiting population, we identified a subset of 58,906 unique individual sampling events (combinations of farm ID, sampling date and individual animal identifier) with paired ELISA and qPCR collected at slaughter, available for the LCA analysis. The apparent seroprevalence and the prevalence of qPCR positivity in the slaughter dataset for farms situated on the South versus the North Island are presented in (Table 3). The raw data for the LCA models are available in S1 Table. Contingency table of the ELISA and qPCR results.

### Test accuracy

Test accuracy results from the Hui-Walter models are presented in Fig 5 and Table 4. The estimated (median of the posterior density distribution) sensitivity of qPCR ranged from 42.3 to 45.2% (SD 2.1 to 2.24) depending on the chosen ELISA threshold, with credible intervals essentially overlapping. The median specificity estimates for qPCR were all close to 99.6% (*SD* < 0.09). The accuracy of ELISA depended on the chosen positivity threshold. At the *SP%* = 90 threshold, the ELISA sensitivity and specificity estimates were 72.8% (CI 68.5—77.4) and

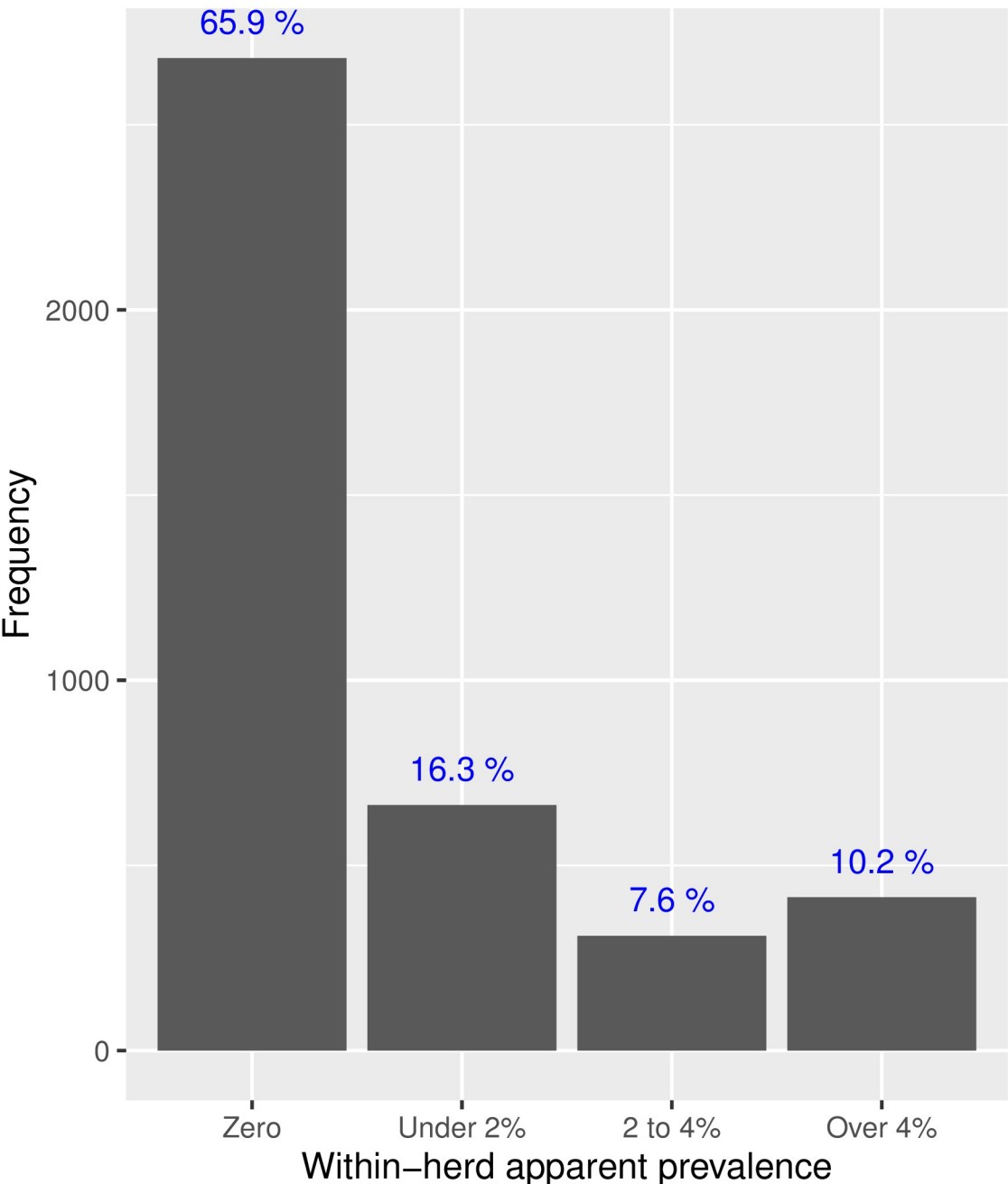

**Fig 2. Proportion of herds in four categories of within-herd prevalence during the first round of on-farm testing (Delimiting population) at the 60 SP% ELISA threshold.**

97.7% (CI 97.3—98.1), respectively. At the 60 threshold, the ELISA sensitivity and specificity estimates were 87.2% (CI 83.4—91.1) and 95.3% (CI 94.8—95.8), respectively.

### True prevalence of active infection

The estimated overall true prevalence of active infection across the two sub-populations was 5.9% (SD = 0.34) for the model at the SP%=60 threshold (and respectively 5.5%, SD = 0.33 for

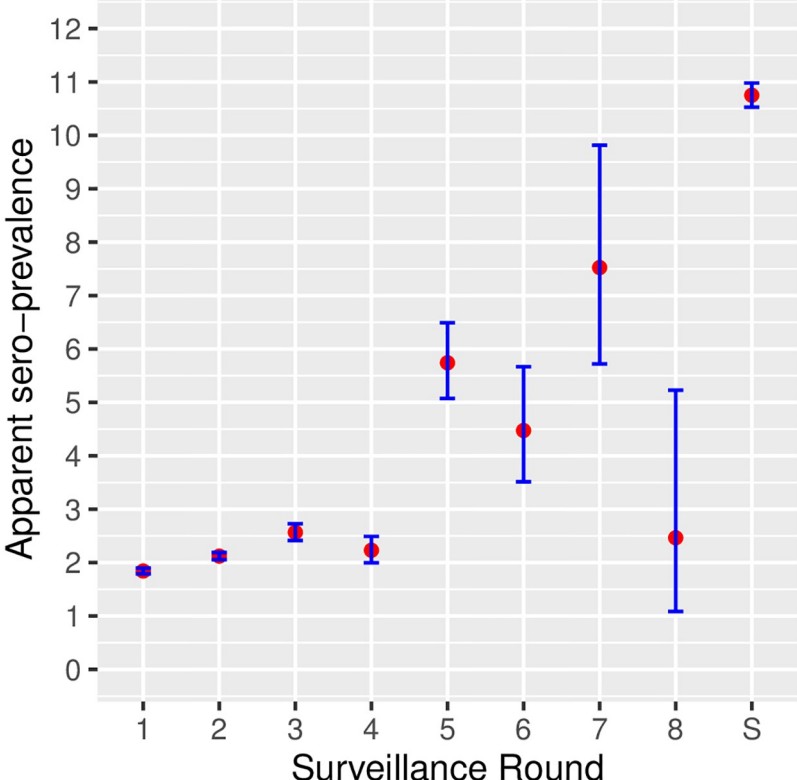

**Fig 3. Overall apparent seroprevalence by sampling round in the Programme (Delimiting population), at threshold SP%=60.** The round denoted "S" represents the slaughter sampling round.

the model at the SP%=90 threshold), see Fig 5. The estimated true prevalence in each sub-population is presented in Table 5.

## Probabilities *Cn* and *Cp* of reporting the qPCR result as a non-dichotomous outcome (free text comment)

The probabilities $Cp = Pr(PCR = C|\text{infected})$ and $Cn = Pr(PCR = C|\text{non- infected})$ were included in the model to gain more insights into test results reported as free text comments "C" and to evaluate if they might represent a bias. Summary measures of posterior probability estimates for Cp and Cn are available in Table 6.

The ratio of probabilities Cp/Cn varied between 5.9 (ELISA threshold SP% = 90) and 8.0 (ELISA threshold SP% = 60). This finding indicated that the laboratory was significantly more prone to not commit to a dichotomous result (positive versus negative) if the sample actually came from a truly infected animal, thus potentially creating a bias in the data. Nevertheless, all the Hui-Walter models were also fitted after omitting the "comment" results and the estimates of all other parameters did not significantly differ (results not shown), likely due to the relative infrequency of qPCR results reported as a comment. The impact of this potential bias was therefore negligible.

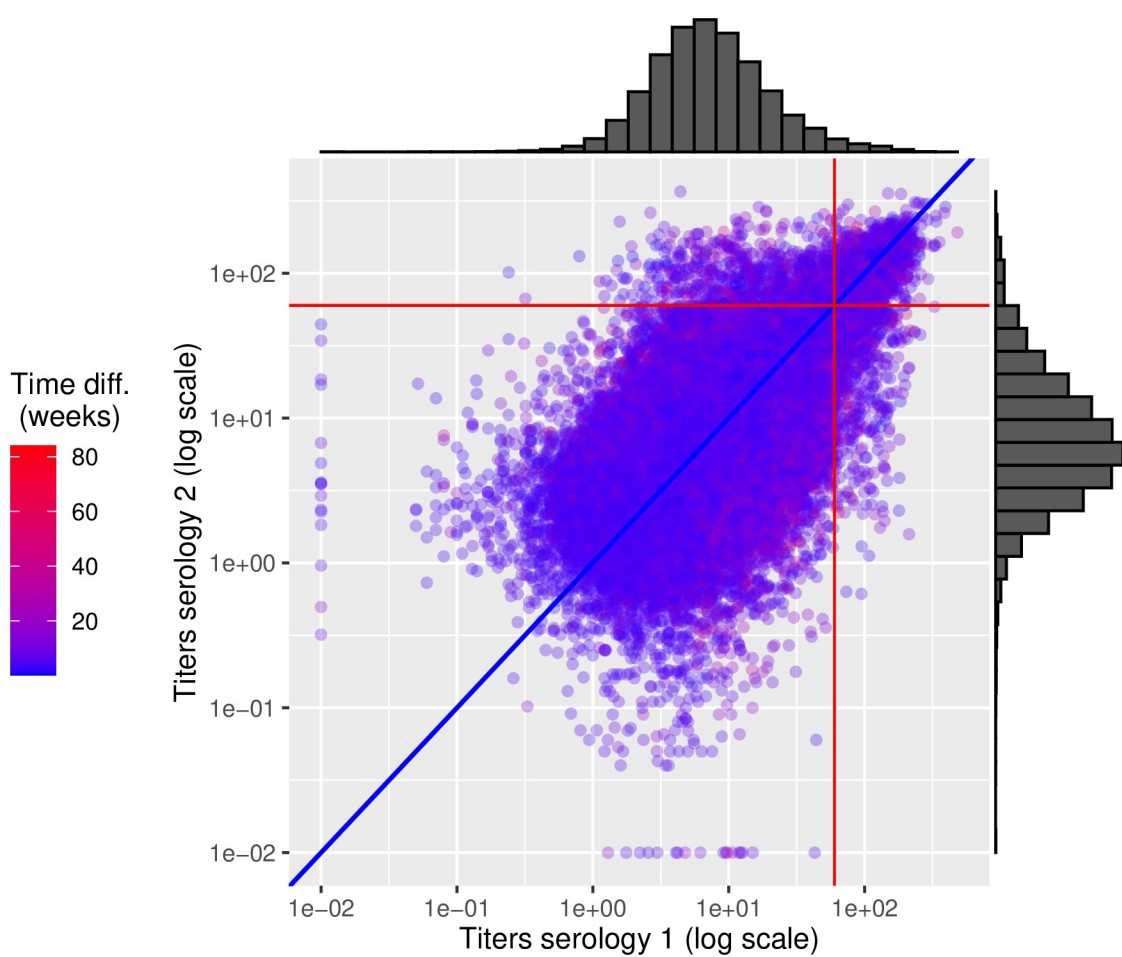

**Fig 4. Correlation in titres between two subsequent rounds in the same animal with marginal titre distribution.** In red the SP% =60 positivity threshold for both distributions, in blue the abline y = x, upper right quadrant T1+T2+ (proportion = 1.64%), upper left quadrant T1-T2+ (proportion = 1.13%), lower right quadrant T1+T2- (proportion = 1.04%), lower left quadrant T1-T2- (proportion = 96.19%).

**Table 3. Apparent seroprevalence (SP% threshold = 60 or 90) and prevalence of positive qPCR for cattle in farms from the North versus the South Island, in the slaughter population.**

| Test | Apparent Prevalence (%) | CI low | CI up |
|---|---|---|---|
| **North island (n = 18949)** | | | |
| ELISA 60 | 6.33 | 5.99 | 6.69 |
| ELISA 90 | 3.54 | 3.28 | 3.82 |
| qPCR | 1.01 | 0.88 | 1.17 |
| **South Island (n = 39554)** | | | |
| ELISA 60 | 11.10 | 10.79 | 11.42 |
| ELISA 90 | 7.41 | 7.15 | 7.67 |
| qPCR | 3.49 | 3.31 | 3.68 |

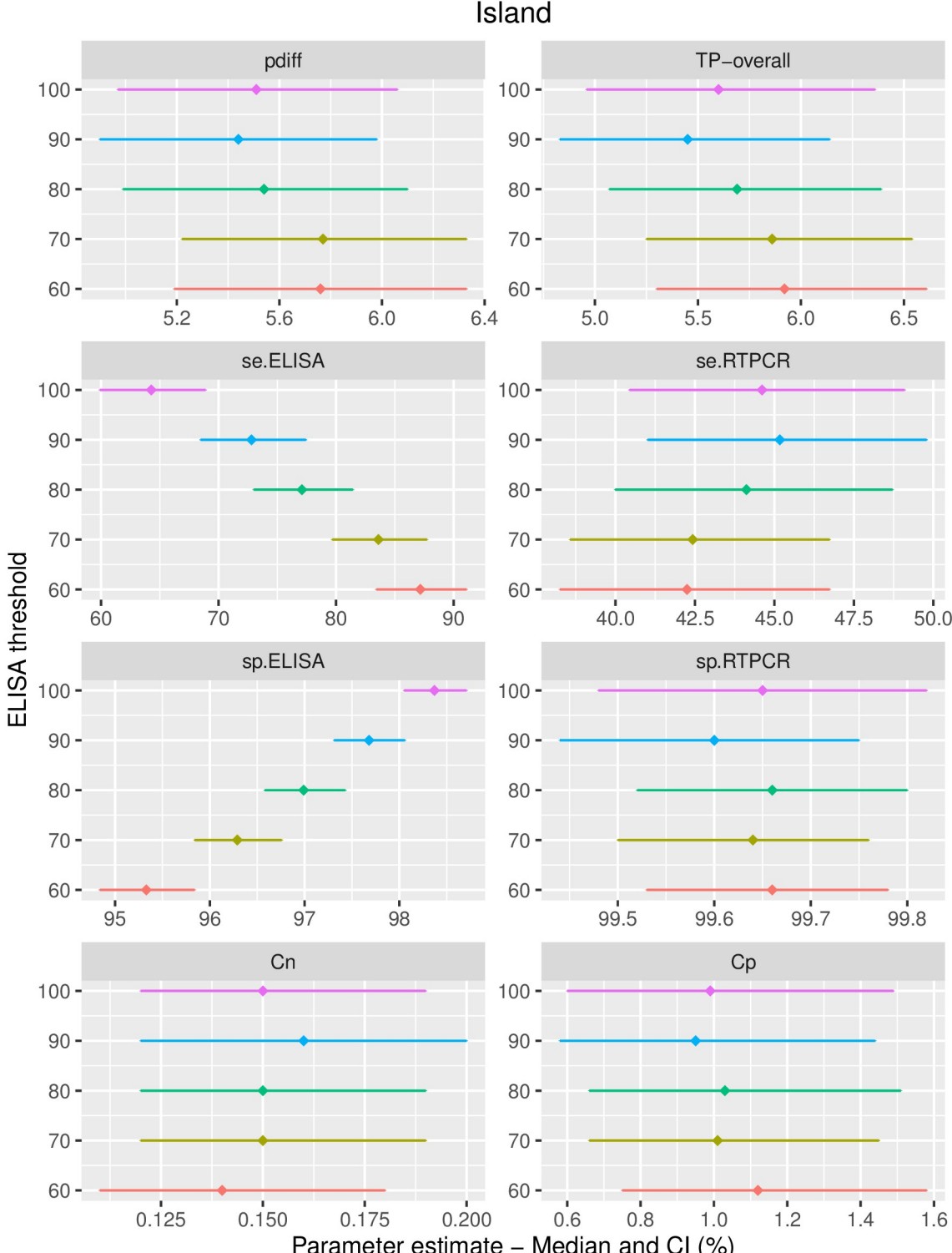

**Fig 5. LCA model results (Median and CI) for varying levels of ELISA positivity threshold.**

**Table 4. Estimated accuracy estimates for ELISA and qPCR for LCA models using ELISA positivity thresholds of SP% = 60 or 90).**

| Parameter | Mean | Median | SD | CI low | CI high |
|---|---|---|---|---|---|
| **ELISA threshold = 60** | | | | | |
| Sensitivity ELISA | 87.19 | 87.17 | 1.95 | 83.42 | 91.11 |
| Sensitivity qPCR | 42.32 | 42.25 | 2.16 | 38.25 | 46.74 |
| Specificity ELISA | 95.33 | 95.33 | 0.26 | 94.84 | 95.84 |
| Specificity qPCR | 99.66 | 99.66 | 0.06 | 99.53 | 99.78 |
| **ELISA threshold = 90** | | | | | |
| Sensitivity ELISA | 72.84 | 72.80 | 2.29 | 68.48 | 77.44 |
| Sensitivity qPCR | 45.23 | 45.17 | 2.24 | 41.01 | 49.79 |
| Specificity ELISA | 97.68 | 97.68 | 0.19 | 97.31 | 98.06 |
| Specificity qPCR | 99.60 | 99.60 | 0.08 | 99.44 | 99.75 |

## Using complementary ELISA screening data to model exposure to Mbovis via constrained priors

The apparent seroprevalence in National Beef Cattle Surveillance at the SP%=60 threshold was 0.18%, two orders of magnitude lower than the observed apparent prevalence in the slaughter dataset (Table 2). Considering that the apparent prevalence is a mixture of true and false positives, this figure would represent the maximum proportion of false positive that could arise if the population was uninfected, i.e. if all cattle positive to ELISA were never exposed to Mbovis.

Corresponding to these estimates, we refitted the LCA model using uniform priors for ELISA specificity between 0.9974 and 1 (and respectively 0.9984 and 1 at the SP%=90 threshold). The estimated adjusted values for test accuracy and true prevalence pertaining to measuring exposure to Mbovis are presented in Table 7.

The difference between this approach and the previous approach modelling "active infection" is presented in Fig 6. By bounding the ELISA specificity using the National Beef Cattle Surveillance dataset (in blue in Fig 6), the sensitivity of both tests became shifted towards lower values, as expected for detecting a broader target condition, such as exposure to Mbovis. However, the estimates for ELISA sensitivity were not significantly different for the various approaches. The posterior distributions for $Sp_{exp}$ for ELISA were dictated by the constrained lower bound used as a specificity prior. The median posterior estimate for $Sp_{exp}$ was thus 99.79% (at the SP%=60 threshold). The corresponding estimated true prevalence of exposure to Mbovis at slaughter was almost double the estimated prevalence of active infection (in blue in Fig 6).

**Table 5. Estimated overall true prevalence of infection for cattle from the North versus the South Island, in models using ELISA positivity thresholds of SP%=60 or 90.**

| ELISA threshold | Mean | Median | CI low | CI high |
|---|---|---|---|---|
| **North Island population** | | | | |
| 60 | 2.03 | 2.02 | 1.60 | 2.55 |
| 90 | 1.78 | 1.76 | 1.35 | 2.29 |
| **South Island population** | | | | |
| 60 | 7.80 | 7.79 | 7.02 | 8.62 |
| 90 | 7.22 | 7.21 | 6.45 | 8.03 |

**Table 6. Summary measures of probabilities Cp and Cn for ELISA positivity thresholds of SP%=60 or 90 (expressed in percentages).**

| ELISA threshold | Mean | Median | CI low | CI high |
|---|---|---|---|---|
| **Cn** | | | | |
| 60 | 0.14 | 0.14 | 0.11 | 0.18 |
| 90 | 0.16 | 0.16 | 0.12 | 0.20 |
| **Cp** | | | | |
| 60 | 1.13 | 1.12 | 0.75 | 1.58 |
| 90 | 0.97 | 0.95 | 0.58 | 1.44 |

## Sensitivity to the priors

We carried out a sensitivity analysis on all the LCA models using flat priors beta(1,1) with random initial values, as well as an alternative set of informative priors. The results did not differ significantly from the models presented for "active infection" with priors presented in Table 1 (results not shown).

## Supplementary analysis for herd level interpretation of the ELISA test

The first scenario favoured the sensitivity of the surveillance strategy at the expense of the specificity. Fig 7 shows that under this scenario, increasing the number of tested animals achieves minimal misclassification of infected groups even for very low prevalence levels, unless the chosen cut-point is above 7%. However, for such values of the cut-point, the surveillance strategy cannot correctly classify un-infected groups due to the lack of specificity of ELISA to detect active infection at the 60 threshold.

The second scenario consisted of targeting prior exposure to Mbovis, using estimates obtained with constrained priors compatible with data from National Beef Cattle Surveillance. This population was virtually unexposed and thus provided a high bound for ELISA specificity estimates. These estimates are thus likely suitable to inform a statement of absence. At the epidemiological threshold (Sp%=90) for ELISA, the specificity of the surveillance strategy was very high, resulting in no false positive diagnostics even at the lowest cut-point, as long as the

**Table 7. Summary measures of test accuracy and true prevalence estimates of "exposure to Mbovis" in the slaughter population, using constrained specificity priors inferred from the National Beef Cattle Surveillance data, for ELISA positivity threshold of SP%=60 or 90.**

| Parameter | Mean | Median | CI low | CI high |
|---|---|---|---|---|
| **ELISA threshold SP% = 60** | | | | |
| se.ELISA | 81.23 | 80.98 | 77.57 | 86.24 |
| se.qPCR | 23.36 | 23.35 | 22.22 | 24.52 |
| sp.ELISA | 99.81 | 99.79 | 99.74 | 99.96 |
| sp.qPCR | 99.87 | 99.88 | 99.70 | 99.99 |
| TP-overall | 11.56 | 11.58 | 10.83 | 12.17 |
| **ELISA threshold SP% = 90** | | | | |
| se.ELISA | 66.22 | 66.04 | 62.72 | 70.68 |
| se.qPCR | 29.74 | 29.74 | 28.20 | 31.29 |
| sp.ELISA | 99.88 | 99.87 | 99.84 | 99.98 |
| sp.qPCR | 99.88 | 99.89 | 99.70 | 99.99 |
| TP-overall | 9.14 | 9.15 | 8.52 | 9.68 |

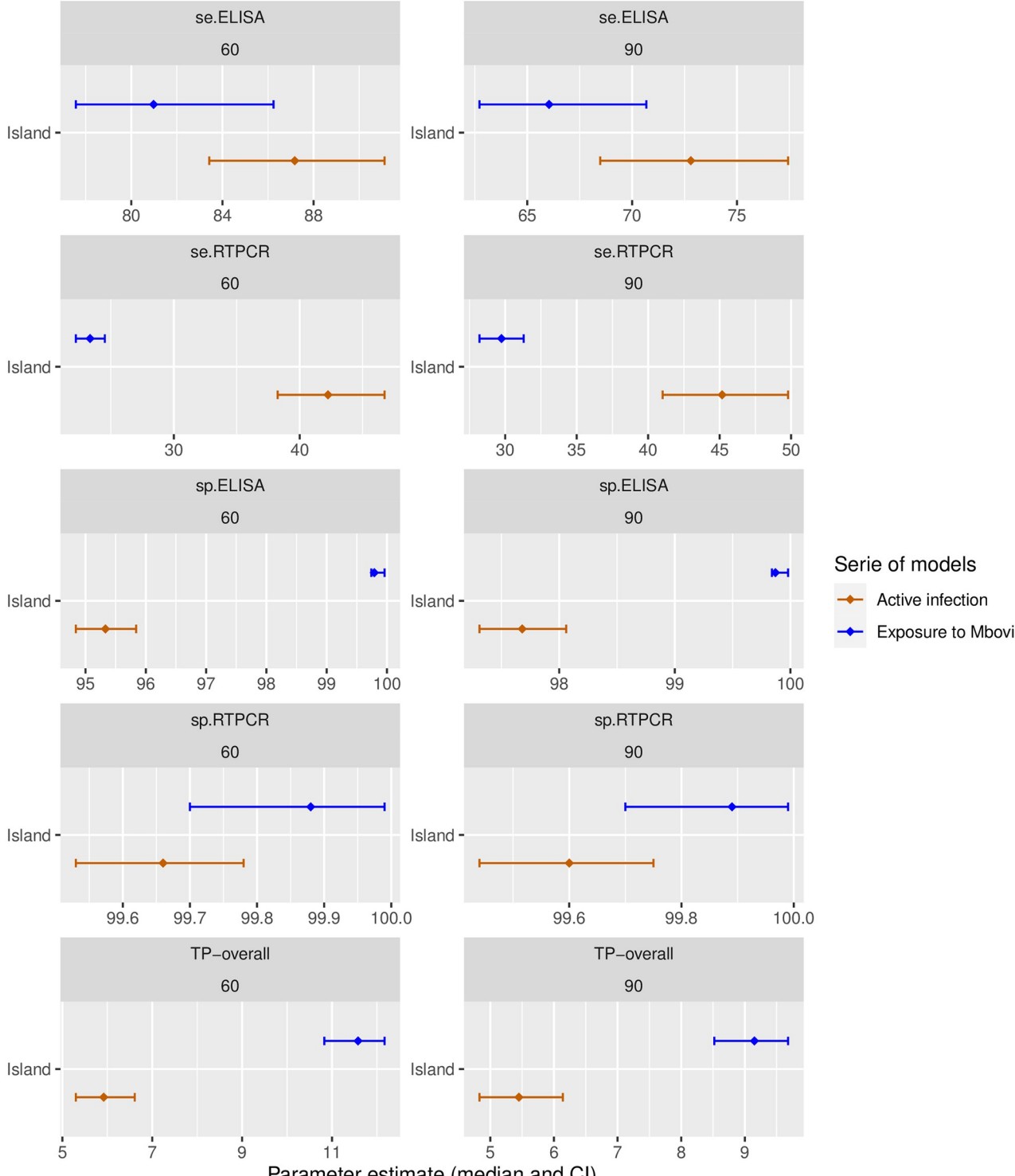

**Fig 6. Summary measures of test accuracy and true prevalence estimates of "exposure to Mbovis" in the slaughter population.** Obtained with constrained ELISA specificity priors using information from National Beef Cattle Surveillance, for ELISA positivity threshold of SP%=60 (left) or 90 (right).

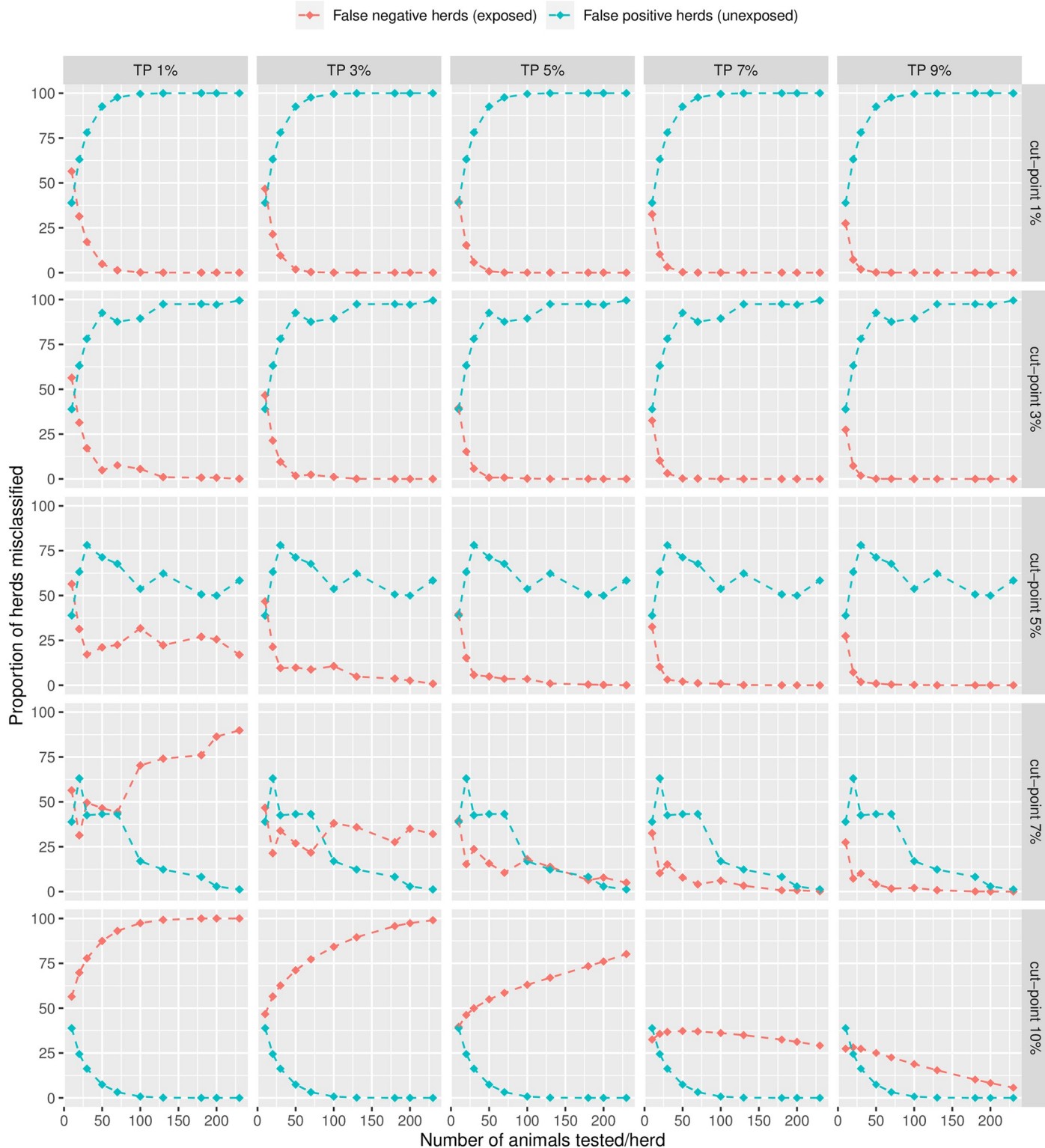

**Fig 7. Proportion of misclassified herds (scenario 1).**

number tested were 200 or more per herd (Fig 8). This was at the expense of the HSe, as not all exposed herds could be detected in low prevalence herds, or at higher cut points.

The jumps observed in Figs 7 and 8 are due to the discrete nature of the tested sample size in the formula. Changing the total herd size to 300 did not significantly alter the above findings.

## Discussion

### Summary of findings

The overall apparent seroprevalence in the Delimiting population (Mbovis EradicationProgramme) increased with successive surveillance rounds, ranging from 1.8% at the first screening round to 10.8% at slaughter. This finding reflected the purposive selection of positive management groups for progression in the surveillance strategy (Fig 3). By contrast, the apparent seroprevalence in the National Beef Cattle Surveillance population was very low, not exceeding 0.26% at the ELISA SP%=60 threshold. Therefore, there is no evidence of widespread infection in the underlying beef cattle population targeted by this surveillance stream.

Fluctuations of ELISA titres between rounds of sampling in the same animals with moderate agreement between subsequent titres could illustrate a lack of reproducibility of the ELISA test at the animal level. Among the 4% of animals likely exposed (at least one test positive at the ELISA SP%=60 threshold), these fluctuations may also in part reflect the dynamics of infection and that of the antibody response to infection over time, with animals at different stages of infection in various sampling rounds. It may also reflect the technical limitations and sensitivity of the test, the impact of which is even more critical in the context of low prevalence of true infection.

Estimates of test accuracy for the serum ELISA and tonsillar swab qPCR were obtained using Bayesian LCA based on paired tests conducted at slaughter. We used the geographical location (farms in the North versus South Island) to partition the data into two sub-populations of distinct prevalences.

The modelled latent condition is dictated by the joint distribution of the ELISA titres and the presence of detectable Mbovis DNA in the palatine tonsillar crypts. We coin the corresponding latent condition "active infection". The results converged towards an ELISA sensitivity estimates of 72.8% with the epidemiological threshold used by the Eradication Programme for serology interpretation (SP%=90). The corresponding qPCR specificity estimate was 99.6%. Estimates for the specificity of the ELISA test were tied to those for the sensitivity of the qPCR test. At the manufacturer's threshold of 60, the qPCR sensitivity was 42.3% (CI 38.3—46.7) and the ELISA specificity was 95.3% (CI 94.8—95.8), and respectively 45.2 (CI 41.0—49.8) and 97.7 (CI 97.3 -98.1) at the epidemiological threshold of 90.

A previous study used a subset of the same data with a gold standard approach to determine individual-level test accuracy and optimal interpretation of ELISA at the herd level [22]. Animal-level sensitivity of the IDvet ELISA test was estimated to be 89.3% (95% CI 87.2—91.5) and the specificity 95% (CI 94.3—95.6), at the threshold of SP%=60. The same report estimated sensitivity of the qPCR to be 38.8%, assuming perfect qPCR specificity. The ELISA sensitivity estimate from this study was likely overestimated considering the use of the qPCR in the gold standard positive definition. The ELISA sensitivity appeared significantly higher than estimates from our analysis. Nevertheless, the sensitivity of the qPCR test and the ELISA specificity were in line with findings from the present study, regarding detecting active infection.

Given the relatively low apparent prevalence of infection in the data, true prevalence estimates were very driven by variations in the specificity of the ELISA test, which appears as an essential parameter to inform the control strategy. The test accuracy estimates derived from

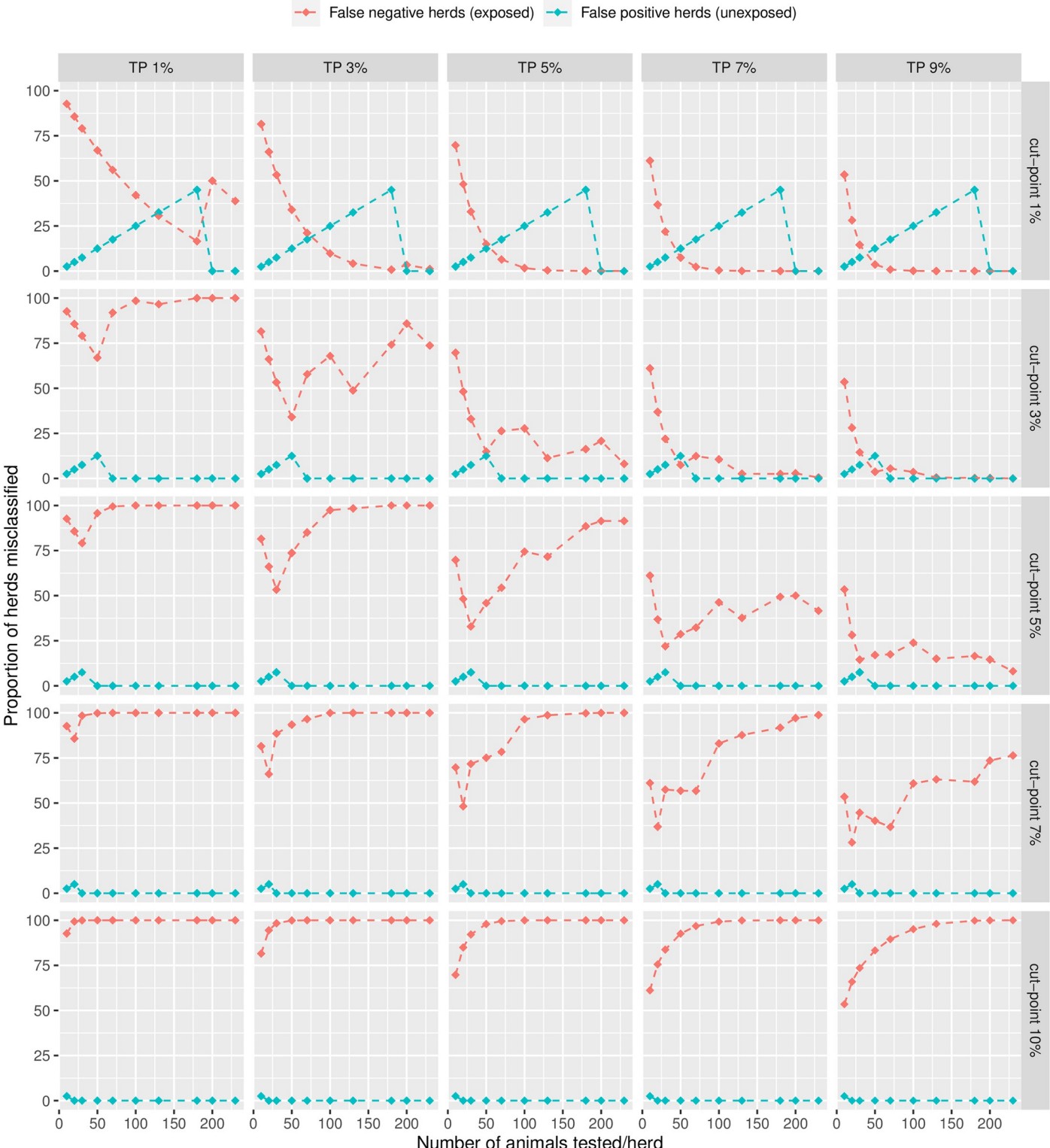

**Fig 8. Proportion of misclassified herds (scenario 2).**

our analyses were further used to explore optimisation of the surveillance strategy (Figs 7 and 8).

## Modelling exposure to Mbovis using a constrained specificity prior

The very low apparent prevalences observed in the National Beef Cattle Surveillance dataset indicate that the specificity of the ELISA test alone was above 99.8%. This finding is discrepant with the ELISA specificity value we obtained by jointly modelling ELISA and qPCR test results using LCA in the slaughter data (95.3% using the SP%=60 threshold). We suggest that this discrepancy largely comes from a shift of the underlying latent condition, determined by the joint response of the test(s) modelled. The ELISA test alone measures exposure to Mbovis, with high specificity. On the other hand, the results of the LCA models pertain to "active infection" due to the joint distribution in the host of the respective biomarkers measured by the two tests used in the LCA. An animal that no longer has detectable infection in its tonsillar crypts but has mounted an immune response following exposure to Mbovis would be considered a false positive in the LCA analysis.

Constrained priors obtained from the National Beef Cattle Surveillance dataset were thus used as an *ad hoc* method to shift the target condition from "active infection" towards "exposure" to Mbovis. This change would adjust, at the same time, for any genuine under-estimation of the specificity estimate potentially associated with selection bias of the slaughter population (see further below).

The National Beef Cattle Surveillance population appeared virtually uninfected, and therefore provided a very precise estimate for $Sp_{exp}$ above 99.8%, representing an upper bound for the ELISA specificity (in blue in Fig 6). The corresponding corrected ELISA sensitivity estimate was 81% at the SP%=60 threshold, and the estimated true prevalence of exposure to Mbovis in the slaughter population was 11.6%. This represents a "best case" scenario for the use of the ELISA test, best suited to screen for Mbovis to demonstrate disease absence. It is important to note that inferences on test accuracy obtained from one population may not be valid when applying the test in a different population, if the distribution of the biomarkers used as a test substrate varies across populations.

## Pathophysiology of Mbovis in cattle and target condition for the testing regimen

At the individual level, titres might correlate poorly with either infection or disease [3, 23], although they may heavily depend on the kit used and more research is required to shed light on the dynamics of antibody response in natural infection with Mbovis. Nevertheless, nasal carriage of Mbovis alone, in the absence of clinical disease, can lead to seroconversion [4]. Similarly, intramammary sub-clinical infections with Mbovis can induce a detectable immune response [24]. These findings suggest that ELISA is a good candidate marker for active infection. By associating ELISA and qPCR on tonsillar swabs using LCA, we were able to model active infection in the absence of a gold standard to detect this condition. Our results demonstrate the value of the IDvet ELISA to detect active infection, with a sensitivity of 87.2% at the lowest positivity threshold (i.e. SP%=60).

The ELISA test being a measure of prior exposure to Mbovis, the host might permanently eliminate infection while keeping serological evidence of past infection. At the herd level, clinical disease caused by Mbovis is often self-limiting. Similarly, some authors describe spontaneous clearance of infection from the herd, which would be more common in small herds [11, 24], presumably due to stochastic fade out due to a lack of contact between infectious and susceptible individuals. These findings, however, need to be interpreted with caution, given the

difficulty to distinguish true clearance from low-level latent infection persisting in the herd. In a recent longitudinal study of 19 recently infected herds followed for two years, clinical mastitis resolved within a couple of months in 88% of the farms. Moreoever, repeated qPCR testing (nasopharyngeal swabs, individual and bulk milk) supported the clearance of infection within the first year in six of the farms [24], although latent infection or lack of sensitivity of the sampling locations could not be ruled out. Nevertheless, the seroprevalence in mixed-aged cows did not decline over time, remaining close to 80% after 1.5 years in all farms, irrespective of the detection of the pathogen. With "active infection" as the latent condition, animals with high titres but recovered from active infection would be considered false positive by ELISA, as can be seen in Fig 1. This observation explains the relative low ELISA specificity obtained in the first series of LCA models targeting "active infection". Animals harbouring the bacteria in their tissues in the absence of shedding may not be at risk of further spreading the organism. However, undetected active infection might remain on the farm, clustered in particular subgroups or age categories. Moreoever, it cannot be ruled out that individuals appearing clear from infection still harbour Mbovis in undetected locations, presenting a potential infection risk in the future.

The current interpretation of ELISA at the herd level, based on a previous DTE report, includes a shift of ELISA positivity threshold towards higher titres (90 instead of 60) and a statistical test to detect higher than expected numbers of animals with very high antibody titres in the herd. This was based on (1) an analysis of the tail of the titre distribution in "gold standard positive" herds, identified by the presence of at least one qPCR positive result and (2) the observation that among all tested animals, higher antibody titres were associated with increased probability of testing qPCR positive. These extra criteria to select infected herds likely result in shifting the target condition towards herds with current active infection, while herds with evidence of exposure might be omitted. Our analysis showed that the two conditions are distinct; targeting one or the other does result in different test accuracy values at the individual level. Given the low prevalence of Mbovis infection, the specificity of the testing regimen is essential in the interpretation of the test results at the herd level for the Programme.

## Other possible bias due to the use of the slaughter population

The slaughter sampling population may not be representative of the screened population that is the target population for the test. This population is biased toward higher prevalence herds, since herds are directed to slaughter upon meeting the case definition. However, the selection process is applied at the herd level and not at the individual-level. The overall apparent seroprevalence at slaughter was 9.7% (CI = 9.4–9.9). With such a low prevalence of infection, we are confident that the slaughter populations represented a real mixture of infected and non-infected animals. The selection bias should therefore have a limited impact for an analysis at the individual-level, with valid estimates so long as the accuracy of the tests do not vary with the prevalence of infection or other latent variables affected by the above selection process. This assumption is standard for this type of analysis.

In the first round of testing the Delimiting population, we observed that positive ELISA results (presumably both true and false positive) tended to cluster within herds (Fig 2). It is natural that infection (true positives) should cluster within herd. But we hypothesise that some herds might also present a higher number of false positive than average, due to chance or other reasons. Such herds might be preferentially selected for slaughter, thus increasing the rate of false positive serological results in the slaughter dataset (selection bias). This bias would cause the specificity of the test to be under-estimated when using these data. It could contribute to the relatively low value of the ELISA specificity estimates in the analysis using slaughter data,

compared to values obtained with constrained priors informed by the National Beef Cattle Surveillance population. This effect remains however limited compared to the effect of shifting the latent condition to "active infection".

Other potential sources of bias for determining test accuracy using the slaughter data could be systematic differences in sample quality due to blood collection *post-mortem*, confounding due to systematic differences in diagnostic laboratory for early rounds of surveillance versus slaughter sampling.

## Validity of the Hui-Walter assumptions and future research

The use of LCA has become a reference method for DTE in the absence of a gold standard [13]. However, this method relies on a set of strict assumptions that may not be realistic in real-world situations. Hui-Walter models have been qualified as being "weakly identifiable" [25], implying that inferences can be very sensitive to modelling assumptions and misspecifications [26]. An array of literature deals with the effect of violating the assumption of conditional independence of the tests and possible remedies [27]. However, we believe this assumption was reasonable here because ELISA and PCR target different substrates (antibody response versus direct detection of Mbovis DNA).

Constant test accuracy across populations is the other main assumptions of LCA models. The issue of violating this assumption was discussed in [28]. Limited simulation work suggests that violating this assumption could lead to a bias towards the sensitivity of the test in the population with a high disease prevalence [28]. We partitioned the cattle population by island to obtain two populations of distinct prevalence. Previous research shows that livestock movements in NZ are highly clustered within island, resulting in similar clustering of infections primarily spreading via livestock movements [19]. These results suggests that the geographic partitioning of population by island was suitable to obtain populations of distinct prevalences, with a higher prevalence expected in the South Island, where cases of disease were first detected and where the majority of cases have subsequently been identified. This assumption was corroborated by estimates of prevalence around 1.8 to 2% in the North Island versus 7.2 to 7.8% in the South Island.

To check the validity of constant test accuracy across both islands, we attempted model diagnostics using internal consistency for different ELISA thresholds. Varying ELISA cutoffs should result in consistent estimates of qPCR accuracy and true prevalences, as can be observed in Fig 5. Our results indicated that splitting the cattle population by North versus South Island likely resulted in sub-populations with even distribution of animal-level attributes potentially influencing the joint distribution of test outcomes. The assumption of constant test accuracy across both islands was therefore reasonable.

We also attempted to partition the cattle population using other covariates representing potential risk factors for Mbovis infection or disease, such as age or main enterprise type (results not shown). Using these alternative sub-populations, estimates of the qPCR accuracy and overall true prevalence varied with varying ELISA thresholds and did not overlap. This finding likely indicated a model mis-specification, suggesting that the sensitivity of the ELISA and/or the qPCR might not be constant across all age groups and enterprise types, so that these appear less suited to partition the cattle population for LCA.

## Conclusion

To conclude, well designed DTE studies involving random sampling in controlled populations are ideally suited for robust DTE; LCA is no surrogate for this approach. However, data from surveillance programmes such as the Mbovis eradication Programme in NZ represent a large

source of information. They can valuably inform DTE [29], although analysis may require extending traditional methods used in DTE or developing specific approaches [30]. Our results confirm that the IDvet ELISA test is an appropriate tool for determining exposure and infection status of herds, both to delimit and confirm the absence of Mbovis.

## Supporting information

**S1 Table. Contingency table of the ELISA and qPCR results.**
(PDF)

## Acknowledgments

We are very grateful to Mary van Andel for facilitating this work and to Emma Bramley for her contribution and support in improving the quality of the collected data and the data flow, which has led to this analysis. We gratefully acknowledge Alexander Crosbie and Cord Heuer for reviewing this manuscript, Doug Begg and Andreas Rohringer for advice about the ELISA test and Diana Jaramillo and Simon Firestone for insightful discussions about the latent class and their help in crafting definitions.

## Author Contributions

**Conceptualization:** Nelly Marquetoux.

**Data curation:** Nelly Marquetoux, Emma Sumner.

**Formal analysis:** Nelly Marquetoux.

**Funding acquisition:** Nelly Marquetoux.

**Methodology:** Nelly Marquetoux, Geoff Jones.

**Supervision:** Matthieu Vignes, Geoff Jones.

**Validation:** Nelly Marquetoux, Matthieu Vignes.

**Visualization:** Nelly Marquetoux.

**Writing – original draft:** Nelly Marquetoux, Matthieu Vignes, Geoff Jones.

**Writing – review & editing:** Nelly Marquetoux, Matthieu Vignes, Amy Burroughs, Kate Sawford, Geoff Jones.

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
