## [Decision Letter · Decision Letter 0]

25 Nov 2022

PONE-D-22-20143Evaluation of the accuracy of the IDvet serological test for Mycoplasma bovis infection in cattle using latent class analysis of paired serum ELISA and quantitative real-time PCR on tonsillar swabs sampled at slaughterPLOS ONE

Dear Dr. Marquetoux,

Thank you for submitting your manuscript to PLOS ONE. After careful consideration, we feel that it has merit but does not fully meet PLOS ONE’s publication criteria as it currently stands. Therefore, we invite you to submit a revised version of the manuscript that addresses the points raised during the review process.

We look forward to receiving your revised manuscript.

Kind regards,

Yong Qi

Academic Editor

PLOS ONE

https://journals.plos.org/plosone/s/fileid=ba62/PLOSOne_formatting_sample_title_authors_affiliations.pdf.

“This work was funded by the Strategic Science Advisory Group on Mbovis (Ministry for 668 Primary Industries). We acknowledge the support provided by the Mbovis Eradication 669 Programme. We are very grateful to Mary van Andel for facilitating this work and to 670 Emma Bramley for her contribution and support in improving the quality of the 671 collected data and the data flow, which has led to this analysis. We gratefully 672 acknowledge Alexander Crosbie and Cord Heuer for reviewing this manuscript, Doug 673 Begg and Andreas Rohringer for advice about the ELISA test and Diana Jaramillo and 674 Simon Firestone for insightful discussions about the latent class and their help in 675 crafting definitions.”

“This work was funded by the Strategic Science Advisory Group on Mycoplasma bovis, Ministry for Primary Industries, New Zealand (https://www.mpi.govt.nz/biosecurity/mycoplasma-bovis/strategic-science-advisory-group/). The funders had no role in study design, data collection and analysis, decision to publish, or preparation of the manuscript.”

Reviewers' comments:

Reviewer's Responses to Questions

**Comments to the Author**

1. Is the manuscript technically sound, and do the data support the conclusions?

Reviewer #1: Yes

Reviewer #2: Yes

2. Has the statistical analysis been performed appropriately and rigorously? 

Reviewer #1: Yes

Reviewer #2: Yes

3. Have the authors made all data underlying the findings in their manuscript fully available?

Reviewer #1: Yes

Reviewer #2: No

4. Is the manuscript presented in an intelligible fashion and written in standard English?

Reviewer #1: Yes

Reviewer #2: Yes

5. Review Comments to the Author

Reviewer #1: The present study is interesting in terms of evaluate the accuracy of the IDvet serological test for Mycoplasma bovis infection. I think authors discuss a lot about the Program, but, if your main goal was to evaluate the IDvet text, I think authors must clearly state if the diagnostic kit works or not in the conclusion section (You added it in the abstract!)

Few considerations:

Line 67 - Do you think mycoplasma can be erradicated? Mycoplasmas are found in animals, insects, plants, etc.

Line 80 - remove "¿"

Reviewer #2: In general terms, the manuscript is well written, the methods are appropriate for the objective of the study. However, there are some moderate and minor comments that need to be addressed the authors.

Moderate comments

• The manuscript should follow the STARD-BLCM standard, and authors must include the STARD-BLCM checklist in the revised version.

• Authors must follow a standard epidemiology structure to define the study population. Define the target population, the source population, eligible population, and sampling population.

• Authors must include Cts cut-off of the qPCR

• Given the sentence in line 222, would be better to run the model with multiple herds instead of two populations based on NI vs SI?

• Beside the seroprevalence results (your data) you do not provide any prior information about differences of Mbovis between NI and Si

• How many herds are in the study (present descriptive stats in term of herds

• Tabla 6, how you can get probabilities over 1?

• The ad-hoc modelling approach should be better explained

• Authors must include the model code

Minor comments

• Lines 42 and 51 must include a reference

• Check line 80

• Check line 258

• Figure 3, 999 rounds?

• Line 393, single sentence paragraph

• Figure 8 is missing from the submission

6. PLOS authors have the option to publish the peer review history of their article (what does this mean?). If published, this will include your full peer review and any attached files.

Reviewer #1: **Yes: **Natália Gaeta

Reviewer #2: No

---

## [Author Response · Author response to Decision Letter 0]

27 Jan 2023

5. Review Comments to the Author

Reviewer #1: The present study is interesting in terms of evaluate the accuracy of the IDvet serological test for Mycoplasma bovis infection. I think authors discuss a lot about the Program, but, if your main goal was to evaluate the IDvet text, I think authors must clearly state if the diagnostic kit works or not in the conclusion section (You added it in the abstract!)

Thanks for a valid comment, which we addressed by adding a sentence at the end of the conclusion. 

Few considerations:

Line 67 - Do you think mycoplasma can be erradicated? Mycoplasmas are found in animals, insects, plants, etc.

The word “eradicate” in the context of the response of NZ to this incursion, specifically refers to clearing the cattle pathogen Mycoplasma bovis (rather than other environmental mycobacteria) from the cattle population of NZ, as indicated - we hope clearly - in the manuscript. Even that may or may not be possible, but it is the aim of the control programme, nevertheless. 

Line 80 - remove "¿" done

Reviewer #2: In general terms, the manuscript is well written, the methods are appropriate for the objective of the study. However, there are some moderate and minor comments that need to be addressed the authors.

Moderate comments

• The manuscript should follow the STARD-BLCM standard, and authors must include the STARD-BLCM checklist in the revised version.

We followed closely the STARD-BLCM standards and the checklist, from the inception of this work. The STARD paper is referenced several times in the manuscript, but in response to this comment we added a clear mention at the very start of the Methods section. Additionally, also in relation to your next question, we have added clarifications and more definitions to the manuscript around the populations, pertaining to the inferences that can be made from our results and the use of the test, in accordance with those guidelines and best practice. However, we feel that adding the actual checklist would be cumbersome and unnecessary, this checklist is readily available from the reference paper for anyone to check for themselves that they are satisfied the manuscript follows these guidelines. We note that our paper is not a methods paper, and the material presented needs to relate to the results of the analyses. We have also consulted experts in DTE about this question, who advised that the checklist typically would not be appended to a DTE manuscript, but exists as a reference. All the inferences, limitations, biases, and departure from assumptions are elaborately discussed as well, and a lot of thoughts around the definition of the latent condition of the LCA are provided in relation with our analyses. We hope this shows appropriate consideration of the concepts central to DTE using LCA, and the mindset of the STARD guidelines. 

• Authors must follow a standard epidemiology structure to define the study population. Define the target population, the source population, eligible population, and sampling population.

We added considerable explanations about these in different places in the MM, including clear definitions, even with some redundancy but with more clarity. Potential biases when extrapolating the results from the study population to the target population are discussed elaborately in the discussion, as well of the effect of the shift of the latent condition in terms of result interpretation, which relate in turn to the appropriate target population that is being considered.

• Authors must include Cts cut-off of the qPCR We did add this information, thank you.

• Given the sentence in line 222, would be better to run the model with multiple herds instead of two populations based on NI vs SI?

No, it would be very difficult (and unnecessary) to run the model while keeping the herd structure, as it would consume a very high number of degrees of freedom by attempting to estimate the within-herd prevalence in each herd, in which we were not interested. We were only interested in interpreting the results of the test at the level of the national herd, for the national eradication programme and future proof of freedom. The sentence line 222 referred more specifically to variable within-herd prevalence of infection reported in the literature, in terms of informing priors, which we translated to relatively diffuse, but plausible, beta priors. 

• Beside the seroprevalence results (your data) you do not provide any prior information about differences of Mbovis between NI and Si

We used the island partition of the population to generate a priori two populations based on distinct prevalences, based on epidemiological considerations important in NZ (clustered movement pattern within island, in general) leading to this biological assumption, as per recommendations for LCA. As stated in the paper, we used independent weakly informative prevalence priors for the animal-level prevalence, with uncertainty encompassing biologically plausible variation between these 2 sub-populations. This fully specifies the prior information we are putting on these prevalences: the prior information on difference in prevalence is implied by this. Since the prevalence priors are equal and independent, this induces an uninformative prior for the difference. We did not want to incorporate prior information about this difference as it would bias the analysis.

• How many herds are in the study (present descriptive stats in term of herds

The answer is 1645, we added that in the descriptive results.

• Tabla 6, how you can get probabilities over 1?

Apologies, all probabilities in the manuscript are expressed in percentage, we added that in the table title of this table, for which it was not obvious. 

• The ad-hoc modelling approach should be better explained

• Authors must include the model code

We included the mathematical equations of the model. The code is only a direct implementation of these equations, in a specific language. If the Editor specifically wants the code, we will provide it in additional material.

Minor comments

• Lines 42 and 51 must include a reference We added relevant references there.

• Check line 80 We fixed the issue

• Check line 258 Corrected

• Figure 3, 999 rounds? We replaced the rounds “999” by the letter “S” and added a note in the title, this represents the slaughter sampling rounds.

• Line 393, single sentence paragraph We corrected that

• Figure 8 is missing from the submission Thanks for notifying us, we uploaded the missing figure

---

## [Decision Letter · Decision Letter 1]

27 Apr 2023

Evaluation of the accuracy of the IDvet serological test for Mycoplasma bovis infection in cattle using latent class analysis of paired serum ELISA and quantitative real-time PCR on tonsillar swabs sampled at slaughter

PONE-D-22-20143R1

Dear Dr. Marquetoux,

We’re pleased to inform you that your manuscript has been judged scientifically suitable for publication and will be formally accepted for publication once it meets all outstanding technical requirements.

Kind regards,

Yong Qi

Academic Editor

PLOS ONE

Additional Editor Comments (optional):

Reviewers' comments:

Reviewer's Responses to Questions

**Comments to the Author**

1. If the authors have adequately addressed your comments raised in a previous round of review and you feel that this manuscript is now acceptable for publication, you may indicate that here to bypass the “Comments to the Author” section, enter your conflict of interest statement in the “Confidential to Editor” section, and submit your "Accept" recommendation.

Reviewer #2: All comments have been addressed

2. Is the manuscript technically sound, and do the data support the conclusions?

Reviewer #2: Yes

3. Has the statistical analysis been performed appropriately and rigorously? 

Reviewer #2: Yes

4. Have the authors made all data underlying the findings in their manuscript fully available?

Reviewer #2: Yes

5. Is the manuscript presented in an intelligible fashion and written in standard English?

Reviewer #2: Yes

6. Review Comments to the Author

Reviewer #2: (No Response)

7. PLOS authors have the option to publish the peer review history of their article (what does this mean?). If published, this will include your full peer review and any attached files.

Reviewer #2: No

---

## [Editor Report · Acceptance letter]

2 May 2023

PONE-D-22-20143R1 

Evaluation of the accuracy of the IDvet serological test for Mycoplasma bovis infection in cattle using latent class analysis of paired serum ELISA and quantitative real-time PCR on tonsillar swabs sampled at slaughter 

Dear Dr. Marquetoux:

I'm pleased to inform you that your manuscript has been deemed suitable for publication in PLOS ONE. Congratulations! Your manuscript is now with our production department. 

Kind regards, 

on behalf of

Dr. Yong Qi 

Academic Editor

PLOS ONE